# Invariant Graph Learning for Treatment Effect Estimation from Networked Observational Data

## ABSTRACT

Treatment effect estimation from networked observational data encounters notable challenges, primarily hidden confounders arising from network structure, or spillover effects that influence unit's outcomes based on neighboring treatment assignments. Existing graph neural network (GNN)-based methods have endeavored to address these challenges, utilizing the GNN's message-passing mechanism to capture hidden confounders or model spillover effects. However, they mainly focus on transductive treatment effect learning on a single networked data, limiting their efficacy in inductive settings for real-world applications where networked data often originates from multiple environments influenced by potentially varying time or geographical regions. In light of this, we introduce the principle of invariance to the task of treatment effect estimation on networked data, culminating in our Invariant Graph Learning (IGL) framework. Specifically, it first generates multiple networked data to simulate diverse environments from a given observational data. Then it further encourages the model to learn environment-invariant representations for confounders and spillover effects. Such a design enables the model to extrapolate beyond a single observed environment, thereby improving the performance of treatment effect estimation in potential new environments. Upon extensive experiments on two real-world datasets, our IGL model demonstrates superior performance compared to state-of-the-art methods.

## KEYWORDS

Networked Data, Spillover Effect, Invariant Learning

## 1 INTRODUCTION

Investigating treatment effects from networked observational data has garnered extensive attention in recent years, due to its potential applications across a wide range of fields including social networks [36, 37], online advertising [44], and financial transactions [4, 14]. While randomized controlled trials (RCTs) remain the gold standard for deriving treatment effects from networked observational data [15, 57], they are often prohibitively expensive, time-consuming, and fraught with ethical complications. Consequently, there is a growing necessity to explore methods for learning treatment effects from networked observational data.

*Conference acronym 'XX, June 03–05, 2018, Woodstock, NY*
© 2018 Association for Computing Machinery.
ACM ISBN 978-x-xxxx-xxxx-x/YY/MM. . . $15.00
https://doi.org/XXXXXXX.XXXXXXX

Estimating treatment effects from networked observational data poses unique challenges compared to traditional independent data due to the following two primary reasons:

- **Presence of interconnected units.** The interconnected units often lead to two main issues, *i.e.,* hidden confounders and spillover effects. Firstly, networked structures frequently contain hidden confounders [11, 16–18]. When estimating treatment effects from networked observational data, it's crucial to consider these biases. A plethora of treatment effect estimation efforts [2, 21, 27, 41, 50, 54, 55] relies on the strong ignorability assumption, which asserts that all confounders are measurable and present within the observed covariates. However, networked data's inherent homogeneity [35] often causes similar units to be interconnected. Consequently, this can introduce hidden confounders that are not apparent within a unit's observable covariates, making the strong ignorability assumption frequently unfeasible. Secondly, the spillover effect is also termed interference [6, 23, 28] or peer effect [24]. It will cause the traditional Stable Unit Treatment Value Assumption (SUTVA) [40] to be invalid. Within networked data, interconnected units imply that a unit's outcome can be swayed by its treatment and by the treatment status of its neighbors. For example, an individual's likelihood of contracting COVID-19 can be heavily influenced by the vaccination status of their immediate contacts.
- **Limitations with observational networked data.** In many practical situations, researchers can access only a limited amount of observational networked data. Moreover, the networked data intended for prediction might differ from the observed set due to reasons like unknown geographic locations or varied collection times. An illustrative example would be gauging an individual's risk of contracting COVID-19 across diverse regions or nations [19, 58]. Consequently, it becomes challenging to learn from observational data and predict treatment effects on unknown or unobserved networked data. This paradigm is commonly referred to as inductive learning [20] or the "out-of-sample" prediction task [41]. Hence, transferring estimators from observed networked data to new data becomes a daunting task, primarily because of the structural variances in different networked data.

To derive treatment effects from networked data, current studies primarily focus on two main aspects: discerning hidden confounders and architecting models to capture spillover effects. A popular strategy, as highlighted by [18], is to harness the message-passing mechanism of graph neural network (GNN) models [20, 25, 48, 53]. Such mechanisms are tailored to encapsulate concealed information within the network structure. More recent endeavors [33] have expanded their focus to tackle unique challenges, encompassing optimization dilemmas [16] or structural disparities [11] inherent in networked data. For modeling spillover effects, recent studies [24, 33, 34] have integrated the treatment status of neighboring nodes as prior information. Together with individual covariates,

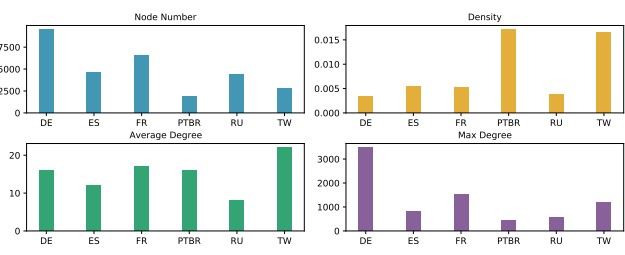

(a) Statistics of node numbers, densities, average and maximum degrees

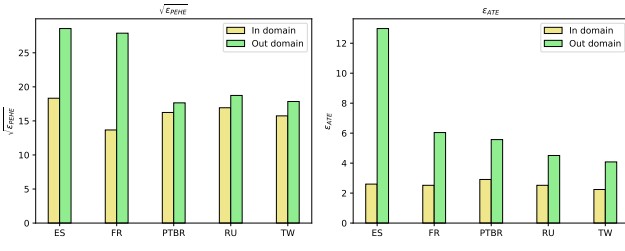

(b) Performance comparison (lower is better)

**Figure 1: Performance of treatment effect estimation based on GNN architecture [25, 33] under two metrics ($\sqrt{\epsilon_{PEHE}}$ and $\epsilon_{ATE}$). (a) Comparison of statistical graph structure properties of different graph data in the Twitch-explicit dataset, including DE, ES, FR, PTBR, RU, and TW domains. (b) "In domain" refers to learning and evaluation on the same graph data, and "Out domain" refers to learning on DE graph data and evaluation on other graph data.**

this information serves as input for the GNN models. It will account for the impact of neighbor treatment assignments on each outcome. Yet, while these techniques have demonstrated efficacy in certain contexts, their applicability tends to be restricted to single-domain graph data. For example, primary applications have been in transductive learning, either within an undivided graph [11, 16, 18, 33] or by segmenting a single graph into discrete subgraphs [24] for distinct learning and evaluation. Notably, when learning and evaluation data come from an identical domain, they inherently possess comparable network topologies and follow the same structural distribution. However, a practical challenge emerges when the evaluation data originates from a disparate domain, potentially leading to structural distribution shifts. As illustrated in Figure 1, networked data from different domains exhibit obvious structural distribution shifts. Although estimators demonstrate commendable performance in learning treatment effects within the given domain, their efficacy tends to diminish in inductive predictions across diverse unseen domains.

A close examination of prevailing methods reveals key limitations. They heavily rely on GNNs to encode network structure information, targeting the capture of confounders, or assimilating neighboring treatments through the message-passing mechanism to learn spillover effects. However, by indiscriminately fitting the observed outcomes in the given networked data, these methods become overly reliant on specific structural information. Consequently, when faced with a structural distribution shift inherent to data from diverse domains, their adaptability falters, leading to a pronounced decline in performance. A logical remedy would involve enabling the model to discern invariant confounders and spillover effects across varied environments. It can ensure the transferability of the treatment effect estimation to alternate domains.

In this paper, we argue that the focus of learning treatment effects should pivot toward capturing these invariant characteristics. To this end, we introduce the Invariant Graph Learning (IGL) framework. Its design intent is to cultivate models that can adeptly estimate treatment effects from networked observational data while preserving cross-domain generalizability. At its core, we employ multiple environment generators, crafting a diverse set of environments from observational networked data. Our objective for confounder learning is to foster a model that discerns invariant confounder representations across these diverse environments. In

terms of spillover effect, we encourage the model to consistently estimate the spillover effect across environments. By ensuring such environmental adaptability, we position our model to seamlessly transition to unseen networked data environments, even when faced with diverse structural distributions. Our key contributions can be summarized as follows:

- We underscore the critical importance of identifying invariant confounders and spillover effects for the task of treatment effect estimation on networked data. It can help the estimators achieve better inductive learning and generalize to diverse domains.
- We propose the Invariant Graph Learning (IGL) framework, tailored to estimate treatment effects from networked observational data. By promoting the capture of invariant confounders and spillover effects, IGL enhances the model's capability to generalize across multiple networked data in new domains.
- We validate the effectiveness of our method on two real-world networked data. Detailed comparisons and in-depth analysis also further confirm the superiority of our approach.

## 2 PRELIMINARIES

In this section, we start with an introduction of the technical preliminaries and then formally present the problem statement of learning individual treatment effects from networked (graph) data.

## 2.1 Notations and Definitions

Firstly, we describe the notations used in this paper. We define $G = (V, E)$ as a graph with node set $V$ and edge set $E$. Let $X = \{x_v | v \in V\} \in \mathbb{R}^{|V| \times d}$ denote the node feature matrix, where $d$ is the feature dimension. We use adjacency matrix $A \in \{0, 1\}^{|V| \times |V|}$ to describe the graph structure, where $A[u, v] = 1$ if edge $(u, v) \in E$, otherwise $A[u, v] = 0$. For a node $v \in V$, let $\mathcal{N}_v = \{u | \text{dis}(v, u) \leq L\}$ denote the $L$-hop neighbors, where $\text{dis}(v, u)$ is the shortest path distance between node $v$ and $u$. The nodes in $\mathcal{N}_v$ and their connections form the ego-graph $G_v$ of node $v$, which is represented as a local node feature matrix $X_v = \{x_u | u \in \mathcal{N}_v\}$ and local adjacency matrix $A_v \in \{0, 1\}^{|\mathcal{N}_v| \times |\mathcal{N}_v|}$. We denote random variables in bold (*e.g.,* **G**) and their corresponding instances in italic font (*e.g., G*). We summarize the detailed notations used in this paper in Table 1. Notice that ego-graphs are not independent samples, but they can

**Table 1: Summary of notations and descriptions.**

| Notation | Description |
|---|---|
| $G, \mathbf{G}$ | graph instance/random variable. |
| $V, E$ | the set of nodes/edges. |
| $A, X$ | the adjacency matrix/node feature matrix. |
| $\mathcal{N}_v$ | the set of $L$-hop neighbors of node $v$. |
| $G_v, \mathbf{G}_v$ | the ego-graph instance/random variable. |
| $A_v, X_v$ | local adjacency matrix/node feature matrix. |
| $x_v, \mathbf{x}_v$ | feature/random variable of feature of node $v$. |
| $T, t_v$ | treatment assignment of all nodes/the node $v$ |
| $T_v$ | treatment assignment of all nodes in $\mathcal{N}_v$. |
| $\mathbf{T}, \mathbf{T}_v, \mathbf{t}_v$ | the random variables of treatment assignment. |
| $Y, y_v$ | observed outcome of all nodes/node $v$. |
| $\mathbf{Y}, \mathbf{y}_v$ | the random variables of observed outcomes. |
| $y_v^1, y_v^0$ | the potential outcomes of the node $v$. |
| $\mathbf{y}_v^1, \mathbf{y}_v^0$ | the random variables of potential outcomes. |
| $(\cdot)_{-v}$ | elements for all nodes in $\mathcal{N}_v$ except the node $v$. |
| $f_{po}$ | the potential outcome function. |
| $\tau_v, \delta_v$ | the ITE/spillover effect of node $v$. |
| $e, \mathcal{E}$ | environment/the support of environments. |
| $\phi_c, \phi_s$ | representation generation functions. |
| $\Psi, \Phi$ | GNN encoder and MLP network. |

be seen as a Markov blanket [22, 51], so that the distribution can be decomposed as $P(\mathbf{G}) = \prod_{v \in V} P(\mathbf{G}_v)$.

In the context of treatment effect estimation, the observational data on the graph $G$ can be denoted as $\{G, T, Y\}$, where $T = \{t_v\}_{v \in V}$ and $Y = \{y_v\}_{v \in V}$ represent treatment assignments and observed outcomes, respectively. We focus on the cases where the treatment variable takes binary values $t \in \{0, 1\}$. Without loss of generality, $t_v = 1$ ($t_v = 0$) means that the node $v$ is under treatment (control). We also let the outcome variable be a scalar and take values on real numbers as $y \in \mathbb{R}$. For ego-graph $G_v$, we define its treatment as $T_v = \{t_v\}_{v \in \mathcal{N}_v}$. And we define $G_{-v}$ and $T_{-v}$ as ego-subgraph and treatment set that include all other nodes and corresponding treatments in $G_v$ and $T_v$ except the central node $v$.

Then we introduce the background knowledge of learning individual treatment effects. To define individual treatment effect (ITE), we start with the definition of potential outcomes on graph data, which is widely adopted in the causal inference literature.

**DEFINITION 2.1 (POTENTIAL OUTCOME ON GRAPH).** *Given the graph $G$, for a node $v$ and its treatment $t_v$, the potential outcome of $v$ under treatment $t_v$, denoted by $y_v^{t_v}$, is defined as the value of $y$ would have taken if the treatment of instance $v$ had been set to $t_v$. The potential outcome of node $v$ can be instantiated via generation function: $y_v^{t_v} = f_{po}(x_v, t_v, G_{-v}, T_{-v})$, where $f_{po}$ can be regarded as a function to generate potential outcome, which takes each unit's treatment assignment, node features, the information (treatment assignments and node features) of its neighbors on the graph.*

As shown in the above definition, the individual's outcome is not only affected by its own feature $x_v$ and treatment $t_v$, but also by neighbor information, *i.e.*, $G_{-v}, T_{-v}$. Hence, the SUTVA assumption is not valid. Below we provide the formal definition of individual treatment effect on graph data.

**DEFINITION 2.2 (INDIVIDUAL TREATMENT EFFECT ON GRAPH).** *Given graph $G = (V, E)$, for each node $v \in V$, the individual treatment effect (ITE) is defined by the difference between the potential outcomes corresponding to $t_v = 1$ and $t_v = 0$:*

$$\tau(\mathbf{G}_v, \mathbf{T}_v) = \mathbb{E}[\mathbf{y}_v^1 - \mathbf{y}_v^0 | \mathbf{x}_v = x_v, \mathbf{G}_{-v} = G_{-v}, \mathbf{T}_{-v} = T_{-v}]$$
$$= \mathbb{E}[f_{po}(\mathbf{x}_v, \mathbf{t}_v = 1, \mathbf{G}_{-v}, \mathbf{T}_{-v}) - f_{po}(\mathbf{x}_v, \mathbf{t}_v = 0, \mathbf{G}_{-v}, \mathbf{T}_{-v})].$$

In this paper, we follow [18, 34], defining ITE as the conditional average treatment effect (CATE). For a given node $v$, ITE is required to keep its own feature and all the states of the surrounding neighbors unchanged to measure the effects of different treatments. For notation simplicity, we define ITE as $\tau_v = \tau(\mathbf{G}_v, \mathbf{T}_v)$. Since the treatments of neighboring nodes also affect ITE, this effect is generally called spillover effect $\delta_v$ in existing literature. Below we give the definition of spillover effect on graph data.

**DEFINITION 2.3 (SPILLOVER EFFECT ON GRAPH).** *The spillover effect of node $v$ under its treatment $t_v$ and its neighbor's treatment assignment $T_{-v}$ is defined as:*

$$\delta_v = \mathbb{E}[f_{po}(\mathbf{x}_v, \mathbf{t}_v, \mathbf{G}_{-v}, \mathbf{T}_{-v}) - f_{po}(\mathbf{x}_v, \mathbf{t}_v, \mathbf{G}_{-v}, 0)].$$

We can observe that the spillover effect is comparing the observed treatments with no treatment from neighbor nodes while keeping all other states unchanged. In this paper, we aim to estimate ITE in the presence of spillover effects in graph data.

## 2.2 Problem Formulation

To estimate the treatment effect on graph $G$, a prevailing approach is to discern the confounders embedded within the network structure. Beyond a node's intrinsic features, a pivotal challenge arises when factoring in the networked structure to uncover these hidden confounders. Several studies [11, 16–18] leverage the message-passing capabilities of GNNs to assimilate the structural intricacies of networked data. By harmonizing both node attributes and structural data, these methodologies adeptly unveil the hidden confounders. Furthermore, to encapsulate the spillover effect, certain strategies [24, 33, 34] integrate the treatment status of neighboring nodes as the prior knowledge and processed via GNNs. The subsequent model parameters are then refined to align with the observed outcomes derived from the networked data. Yet, a significant portion of these methods operate within the graph data in a single domain. Despite their efficacy, real-world applications often necessitate predicting treatment effects on novel networks devoid of observed outcomes. This task is commonly termed as out-of-sample estimation or inductive learning. Now we provide a detailed definition of inductive learning tailored for multi-graph data scenarios.

**DEFINITION 2.4 (INDUCTIVE TREATMENT EFFECT LEARNING ACROSS GRAPHS).** *Given two graphs $G$ and $G'$, assume they follow the same treatment and outcome generation mechanism. A potential outcome estimator $\hat{f}_{po}$ can be learned from $G$ with its observational treatments $T$ and outcomes $Y$. Inductive treatment effect learning requires that for the new graph data $G'$ with its treatments $T'$, the estimator $\hat{f}_{po}$ can also correctly predict the potential outcome $Y'$.*

Let $\mathcal{E}$ represent the support of environments. We posit that the entire graph is formulated via $G \sim p(\mathbf{G}|\mathbf{e})$, where $\mathbf{e}$ signifies a latent environmental variable influencing data distribution. Graph

data derived from varied environments might result in distinct distributions of node features or overall graph structures. If graphs $G$ and $G'$ are sampled from an identical environment, denoted as $G, G' \sim p(\mathbf{G}|\mathbf{e} = e), e \in \mathcal{E}$ (implying minimal distribution shifts in node features or graph structures), then the estimator $\hat{f}_{po}$ transitioned to the new graph $G'$ should yield commendable performance. However, this situation is not always true in practice. New graph data might stem from disparate environments due to variables like undisclosed geographic locations or variances in collection periods. A practical illustration could be assessing an individual's susceptibility to COVID-19 across distinct regions, countries, or timelines [19, 58]. Moreover, complications arising from hidden confounders and the spillover effect might exacerbate the performance downturn of estimators predominantly built on GNNs. As depicted in Figure 1, there's an anticipated divergence in structural attributes (*e.g.*, edge density or node degree) between training and evaluation graphs. This divergence often leads to suboptimal performance. Such challenges underscore the intricacies of inductive potential outcome predictions spanning multiple graphs.

## 2.3 Theoretical Analysis

While the graph data intended for prediction might originate from diverse environments, the principles of invariant learning [3, 8, 38] offer a pathway to achieve cross-environment generalization. Guided by the existing literature on invariant learning, we put forth the following assumption.

ASSUMPTION 2.1 (INVARIANT POTENTIAL OUTCOME GENERATION). *Given ego-graph $\mathbf{G}_v$ and treatment $\mathbf{T}_v$, there exist invariant representation generators $\phi_c$ and $\phi_s$. They can generate representations: $\mathbf{c}_v = \phi_c(\mathbf{G}_v)$ and $\mathbf{s}_v = \phi_s(\mathbf{G}_{-v}, \mathbf{T}_{-v})$. And they satisfy the following properties: i) Sufficiency condition: $\mathbf{y}_v = \hat{f}_{po}(\mathbf{t}_v, \mathbf{c}_v, \mathbf{s}_v) + \epsilon$, where $\epsilon$ signifies an independent noise. ii) Invariance condition: $\forall e, e' \in \mathcal{E}$, $p_e(\mathbf{y}_v|\mathbf{t}_v, \mathbf{c}_v, \mathbf{s}_v) = p_{e'}(\mathbf{y}_v|\mathbf{t}_v, \mathbf{c}_v, \mathbf{s}_v)$.*

Assumption 2.1 posits the existence of representations in the potential outcome generation process that maintain an invariant relationship with potential outcomes across different environments. Specifically, there exists an environment-invariant representation $\mathbf{c}_v$ in $\mathbf{G}_v$, and an invariant spillover effect representation $\mathbf{s}_v$ in $\mathbf{G}_v$ and $\mathbf{T}_v$. The sufficiency conditions suggest that $\mathbf{t}_v$, $\mathbf{c}_v$, and $\mathbf{s}_v$ are adept at faithfully representing the original observational data. Moreover, $\mathbf{y}_v$ can be formulated based on these through the outcome estimation function $\hat{f}_{po}$. As for the invariance condition, it asserts that $\mathbf{t}_v$, $\mathbf{c}_v$, and $\mathbf{s}_v$ retain a cross-environmental consistency in their relationship with the outcome $\mathbf{y}_v$. This enduring relationship underpins the solvability of the problem as delineated in Definition 2.4. Subsequently, we adapt the widely acknowledged unconfoundedness assumption [40] on ego-graph to align with our settings, leading to our second assumption.

ASSUMPTION 2.2 (UNCONFOUNDEDNESS ON EGO-GRAPH). *For any node $v$, given the ego-graph $\mathbf{G}_v$, the potential outcomes are independent with the treatment assignments, i.e., $\mathbf{y}_v^1, \mathbf{y}_v^0 \perp\!\!\!\perp \mathbf{t}_v, \mathbf{T}_{-v}|\mathbf{G}_v$.*

Now we give a brief proof of the identification of potential outcome in Definition 2.4. Given graph $G \sim p(\mathbf{G}|\mathbf{e} = e)$ and $G' \sim p(\mathbf{G}|\mathbf{e} = e')$, we define their treatments and outcomes as $\{T, Y\}$ and $\{T', Y'\}$, respectively. Based on the above assumptions,

across-environment identification of the expectation of potential outcomes $\mathbf{y}_v^1$ and $\mathbf{y}_v^0$ can be proved. Here we take $\mathbf{y}_v^1$ as an example.

$$\mathbb{E}[\mathbf{y}_v^1|\mathbf{t}_v = 1, \mathbf{x}_v = x_v', \mathbf{G}_{-v} = G_{-v}', \mathbf{T}_{-v} = T_{-v}'] \quad (1)$$

$$= \mathbb{E}[f_{po}(\mathbf{t}_v = 1, \mathbf{x}_v = x_v', \mathbf{G}_{-v} = G_{-v}', \mathbf{T}_{-v} = T_{-v}')] \quad (2)$$

$$= \mathbb{E}[\hat{f}_{po}(\mathbf{t}_v = 1, \mathbf{c}_v = \phi_c(G_v'), \mathbf{s}_v = \phi_s(G_{-v}', T_{-v}')] \quad (3)$$

$$= \mathbb{E}[\hat{f}_{po}(\mathbf{t}_v = 1, \mathbf{c}_v = \phi_c(G_v), \mathbf{s}_v = \phi_s(G_{-v}, T_{-v})] \quad (4)$$

$$= \mathbb{E}[\hat{f}_{po}(\mathbf{t}_v = 1, \mathbf{c}_v = \phi_c(G_v), \mathbf{s}_v = \phi_s(G_{-v}, T_{-v})|\mathbf{c}_v = \phi_c(G_v)] \quad (5)$$

$$= \mathbb{E}[\hat{f}_{po}(\mathbf{t}_v = 1, \mathbf{c}_v = \phi_c(G_v), \mathbf{s}_v = \phi_s(G_{-v}, T_{-v})|\mathbf{t}_v = 1, \quad (6)$$
$$\mathbf{c}_v = \phi_c(G_v), \mathbf{s}_v = \phi_s(G_{-v}, T_{-v})]$$

$$= \mathbb{E}[\mathbf{y}_v|\mathbf{t}_v = 1, \mathbf{c}_v = \phi_c(G_v), \mathbf{s}_v = \phi_s(G_{-v}, T_{-v})]. \quad (7)$$

Here, the equation (2) is based on the definition of potential outcome in this setting; equations (3) and (4) are inferred from Assumption 2.1; equation (5) is a straightforward derivation; equation (6) is based on Assumption 2.2; and equation (7) is based on the widely used consistency assumption [40]. Based on the above proof for the identification of potential outcomes, the identification of ITE can be straightforwardly derived.

## 3 METHODOLOGY

In this section, we introduce Invariant Graph Learning (IGL) to solve the problem of inductive potential outcome learning across graphs. The overview of IGL is depicted in Figure 2, which consists of four components, including environment generator, invariant confounder and spillover effect learning, and outcome predictor.

### 3.1 Environment Generator

For a given graph data $G = (A, X)$, we first need to simulate diverse environments. We define the environment generators as $\{g_{\theta_k}\}_{k=1}^{K}$ with parameters $\{\theta_1, ..., \theta_K\}$. Specifically, the environment generation process is defined as $G^k = g_{\theta_k}(G) = (A^k, X^k)$. For the node features $X$ and graph structures $A$, we model the environment change as an additive function that injects perturbations, *i.e.*, $X^k = X + \Delta_X^k$ and $A^k = A \oplus \Delta_A^k$, where $\oplus$ means the element-wise exclusive OR operation and $\Delta_A^k \in \{0, 1\}^{|V| \times |V|}$ is a binary matrix. We treat $\Delta_X^k$ as trainable parameters in $\theta_k$. To generate $\Delta_A$, each edge will be associated with a random variable $p_e \sim \text{Bernoulli}(\omega_e)$, where the edge exists if $p_e = 1$ and is dropped otherwise. We parameterize the Bernoulli weight $\omega_e$ by leveraging a GNN encoder $\Psi$ and a MLP network $\Phi$:

$$\{h_1, ..., h_{|V|}\} = \Psi(G), \quad \omega_e = \Phi([h_v, h_u]), \quad (8)$$

where $\{h_v\}_{v \in V}$ denotes the node representations. To train the model in an end-to-end fashion, we relax the discrete $p_e$ to be a continuous variable in $[0, 1]$ and utilize the Gumbel-Max reparametrization trick. Specifically, $p_e = \text{Sigmoid}((\log \delta - \log(1 - \delta) + \omega_e)/\tau)$, where $\delta \sim \text{Uniform}(0,1)$. As the temperature hyper-parameter $\tau \to 0$, $p_e$ gets close to the being binary. Hence, we can generate $K$ viewed graphs $\{G^1, ..., G^K\}$ to simulate $K$ different environments.

### 3.2 Learning Invariant Confounder

For the given graph $G$, we first encode the node features $X$ and graph structures $A$ into node representations via a GNN encoder,

**Figure 2: The overview of the proposed Invariant Graph Learning (IGL) for Treatment Effect Estimation.**

*i.e.,* $H = \Psi_c(A, X)$, which is expected to capture all potential confounders. Given the generated graphs from $K$ environments, we can obtain $K$ groups of node representations, *i.e.,* $\{H^1, ..., H^K\}$. According to Assumption 2.1, to learn invariant confounders, we need to extract those "environment-invariant" representations from the captured confounders. Hence, we adopt the idea of contrastive learning [9, 56] to ensure that the learned confounder representation is invariant across environments. We define positive pairs as the same instance across different environments and negative pairs as other different instances. Specifically, given an anchor instance $h_v \in H^k$, we randomly sample an environment index $k'$ and choose the same instance in this environment $h'_v \in H^{k'}$ as a positive instance, and we randomly sample another $M$ instance $\widetilde{h_u}$ with $v \neq u$ from any environment as a negative sample. We can define the following invariant contrastive learning objective as:

$$\mathcal{L}_{con}^k = \frac{1}{|V|} \sum_{v \in V} -\log \frac{\exp(\text{sim}(h_v, h'_v)/\tau_c)}{\sum_{u=1, u \neq v}^{M} \exp(\text{sim}(h_v, \widetilde{h_u})/\tau_c)}, \quad (9)$$

where $\text{sim}(h_v, h_u) = h_v^\top h_u / \|h_v\|\|h_u\|$ is the similarity function and $\tau_c$ is the temperature coefficient.

### 3.3 Learning Invariant Spillover Effect

We now introduce the modeling process for spillover effects. According to Assumption 2.1, we need to design additional modules to learn spillover effects that are invariant across environments. For the given confounders, treatments, and graph structure, we propagate the treatment assignment and confounder representations with the GNN module. Specifically, we define $S^k = \{h_v \odot t_v\}_{v \in V}$, where $\odot$ is the broadcasted element-wise product. Then we adopt a GNN encoder $\Psi_s$ to generate the representations of spillover effects: $Z^k = \Psi_s(A^k, S^k)$. Similarly, to encourage the learned representations of spillover effects to be invariant across $K$ environments, we also employ the invariant contrastive learning objective:

$$\mathcal{L}_{sp}^k = \frac{1}{|V|} \sum_{v \in V} -\log \frac{\exp(\text{sim}(z_v, z'_v)/\tau_s)}{\sum_{u=1, v \neq u}^{M} \exp(\text{sim}(z_v, \widetilde{z_u})/\tau_s)}. \quad (10)$$

The estimation of treatment effect may be biased due to possible distribution discrepancy between treatment and control groups.

Therefore, we use the strategy of representation balancing to add a regularization term on representation learning for confounder and spillover effect. In our implementation, we follow [41] and employ Wasserstein-1 distance to achieve representation balancing.

### 3.4 Optimization Objective

To predict the outcome for node $v$, we take the representations $h_v^k$, $z_v^k$, and the treatment $t_v$ as input to the outcome prediction function $f_{out}$:

$$f_{out}(h_v^k, z_v^k, t_v) = \begin{cases} f_1([h_v^k \| z_v^k]) & \text{if } t_v = 1 \\ f_0([h_v^k \| z_v^k]) & \text{if } t_v = 0 \end{cases}, \quad (11)$$

where $f_0$ and $f_1$ are learnable functions and we implement them with MLPs. Hence, we can obtain the estimated outcome $\hat{y}_v^k$ for node $v$ under the environment $k$. Then we define the following outcome learning objective:

$$\mathcal{L}_{out}^k = \frac{1}{|V|} \sum_{v \in V} (y_v - \hat{y}_v^k)^2 + \gamma \mathcal{L}_b^k + \alpha(\mathcal{L}_{con}^k + \mathcal{L}_{sp}^k), \quad (12)$$

where the first item is the Mean Squared Error (MSE) loss; $\mathcal{L}_b^k$ is the representation balancing loss; the last item is the contrastive learning loss for invariant constraint; $\gamma$ and $\alpha$ are two hyperparameters. To achieve invariant learning across diverse environments, we encourage the model to optimize the mean and variance over $K$ environments. Furthermore, the environment generators should also explore challenging environments to enhance the generalization of the model to unseen environments. Hence, we define the min-max optimization objective:

$$\min_{\Theta} \left\{ \frac{1}{K} \sum_{k=1}^{K} \mathcal{L}_{out}^k + \beta \cdot \text{Var}(\{\mathcal{L}_{con}^k + \mathcal{L}_{sp}^k : 1 \leq k \leq K\}) \right\}, \quad (13)$$

$$s.t. \, \theta_1^*, ..., \theta_K^* = \arg \max_{\theta_1, ..., \theta_K} \text{Var}(\{\mathcal{L}_{con}^k + \mathcal{L}_{sp}^k : 1 \leq k \leq K\}),$$

where $\Theta$ represents the parameters of modules $\Psi_c, \Psi_s, f_{out}$, $\text{Var}(\cdot)$ represents the variance for a series of losses, and $\beta$ represents the balancing coefficient. We refer to the above training framework as

---

**Algorithm 1:** IGL for Treatment Effect Estimation

---

**Input:** Observational networked data $G = (A, X)$, treatment $T$ and outcome $Y$; Initialized parameters $\Theta$ including networks of $\Psi_c$, $\Psi_s$, $f_{out}$; Initialized parameters $\theta = \{\theta_k\}_{k=1}^K$ of generators $\{g_{\theta_k}\}_{k=1}^K$; Learning rates $\eta_1$ and $\eta_2$.

**Output:** The trained parameters.

1: **while** not converge or maximum epochs not reached **do**
2:     **for** $i = 1, ..., N$ **do**
3:        Obtain augmented graphs $\{G^k \leftarrow g_{\theta_k}(G)\}_{k=1}^K$;
4:        Compute the loss for confounder $\{\mathcal{L}_{con}^k\}_{k=1}^K$;
5:        Compute the loss for spillover effect $\{\mathcal{L}_{sp}^k\}_{k=1}^K$;
6:        Compute $J_1(\theta) \leftarrow \text{Var}(\{\mathcal{L}_{con}^k + \mathcal{L}_{sp}^k : 1 \le k \le K\})$;
7:        Update $\theta_k \leftarrow \theta_k + \eta_1 \nabla_{\theta_k} J_1(\theta), k = 1, ..., K$;
8:        **if** $i == N$ **then**
9:           Compute $J_2(\Theta) \leftarrow \frac{1}{K} \sum_{k=1}^K \mathcal{L}_{out}^k + \beta J_1(\theta)$;
10:          $\Theta \leftarrow \Theta - \eta_2 \nabla_\Theta J_2(\Theta)$;
11:        **end if**
12:     **end for**
13: **end while**

---

Invariant Graph Learning (IGL). We provide the detailed training process in Algorithm 1.

# 4 EXPERIMENTS

In this section, we verify the superiority of the proposed framework through experiments. Specifically, our purpose is to verify the following aspects: (1) Performance comparison of our method with existing methods. (2) Effects of environment generator and invariant learning modules on performance. (3) Sensitivity of different hyperparameters to the final performance.

## 4.1 Dataset and Simulation

*4.1.1* ***Dataset Details***. We adopt two real-world social network datasets Twitch-explicit and Facebook-100 from [29]. In both datasets, a node is a user and an edge denotes their relationship. Additional details about each dataset are provided as follows.

- **Twitch-Explicit** [39] includes seven networked data where nodes represent Twitch users and edges represent their mutual friendships. Each networked data is collected from a particular region, including DE, ENGB, ES, FR, PTBR, RU, and TW. Due to regional differences, these networks have obvious differences in graph structure, as shown in Figure 1(a). We choose DE for training, ENGB for validation, and ES, FR, and PTBR for testing.
- **Facebook-100** [46] includes 100 Facebook friendship network snapshots from the year 2005, and each network contains nodes as Facebook users from a specific American university. We adopt eight networks in our experiments: John Hopkins, Caltech, Amherst, Cornell, Yale, Penn, Brown, and Texas. We use Penn, Brown, and Texas for testing, Cornell and Yale for validation, and use three remaining graphs for training. These graphs have significantly diverse sizes, densities, and degree distributions.

In both datasets, the raw node features are high-dimensional and very sparse, following studies [17, 24, 31], we use LDA [7] to reduce the node feature dimension to 50.

*4.1.2* ***Treatment and Potential Outcome Simulations***. Since the counterfactual outcomes are hard to obtain, we follow the standard practice in the existing literature [24, 33] to manually generate treatments and outcomes. Given the node feature $x_v$, the treatment $t_v$ for node $v$ is generated as $t_v \sim \text{Bernoulli}(\text{Sigmoid}(x_v w_v))$, where $w_v \in \mathbb{R}^d$ is a vector that each element follows a Gaussian distribution. Given the ego-graph $G_v$ and its treatment assignment $T_v$, the potential outcome of node $v$ is generated by

$$y_v = g_0(x_v) + \lambda_t g_t(x_v, t_v) + \lambda_s g_s(G_{-v}, T_{-v}) + \epsilon_v, \quad (14)$$

where $\lambda_t$ and $\lambda_s$ are strengths to ITE and spillover effect. We define $g_0(x_v) = w_0 x_v$ ($w_0 \sim \mathcal{N}(0, I), w_0 \in \mathbb{R}^d$) as the outcome of instance $v$ when $t_v = 0$ and without network interference. $g_t(\cdot)$ and $g_s(\cdot)$ summarize the ITE and spillover effect, respectively. $\epsilon_v \sim \mathcal{N}(0, 1)$ denotes the Gaussian noise. And we specify $g_t(\cdot)$ and $g_s(\cdot)$ as

$$g_t(x_v, t_v) = t_i \cdot (w_t x_v + \epsilon), \quad (15)$$

$$g_s(G_{-v}, T_{-v}) = \sigma\left(\frac{1}{|\mathcal{N}_{-v}|} \sum_{u \in \mathcal{N}_{-v}} t_u \cdot g_t(x_u, t_u)\right). \quad (16)$$

## 4.2 Experiment Settings

*4.2.1* ***Metrics***. We adopt two metrics: Rooted Precision in Estimation of Heterogeneous Effect $\sqrt{\epsilon_{PEHE}}$ and Mean Absolute Error $\epsilon_{ATE}$. These metrics can be defined as follows:

$$\sqrt{\epsilon_{PEHE}} = \sqrt{\frac{1}{|V|} \sum_{v \in V} (\tau_v - \hat{\tau}_v)^2}, \; \epsilon_{ATE} = \left| \frac{1}{|V|} \sum_{v \in V} (\tau_v - \hat{\tau}_v) \right|, \quad (17)$$

where $\hat{\tau}_v$ is the estimation and $\tau_v$ is ground-truth. Lower is better for both metrics.

*4.2.2* ***Baselines***. To investigate the superiority of the proposed IGL, we compare it with the following three categories of baselines.

- **Traditional methods.** These methods are classic algorithms for ITE estimation and do not consider the structural information of the graph data, including linear regression (LR), Treatment-agnostic Representation Networks (TARNet), and Counterfactual Regression (CFR) [41].
- **Considering the graph structures.** Netdeconf [18] uses GCN [25] as the backbone model to capture hidden confounders, but does not consider the spillover effect on the graph.
- **Considering the spillover effect on graphs.** GNN-HSIC [34], GCN-HSIC [34], NetEst [24] and HyperSCI [33] perform ITE estimation on the graph and also consider the spillover effect. For HyperSCI, we replaced their hypergraph modules with GNN layers, which can be applied to ordinary graphs.

*4.2.3* ***Implementation details***. The configuration details of our model are as follows. For the environment generator, confounder, and spillover effect learning modules, we use the GCN [25] or GAT [48] layer as the encoder. For the representation dimension of the confounder and spillover effect, we set it to 64. Our other hyperparameter settings are: $K = 3, \eta_1, \eta_2 = 1e - 4, N = 1, \alpha = 1, \beta = 1, \gamma = 0.5$. For all experimental results, we perform 10 random runs and report the mean and standard derivations.

**Table 2: Performance comparison of ITE esitmation on Twitch-Explicit and Facebook-100 datasets.**

| | Twitch-Explicit | | | | | | Fackbook-100 | | | | | |
|---|---|---|---|---|---|---|---|---|---|---|---|---|
| | ES | | FR | | PTBR | | Penn | | Brown | | Texas | |
| Method | $\sqrt{\epsilon_{PEHE}}$ | $\epsilon_{ATE}$ | $\sqrt{\epsilon_{PEHE}}$ | $\epsilon_{ATE}$ | $\sqrt{\epsilon_{PEHE}}$ | $\epsilon_{ATE}$ | $\sqrt{\epsilon_{PEHE}}$ | $\epsilon_{ATE}$ | $\sqrt{\epsilon_{PEHE}}$ | $\epsilon_{ATE}$ | $\sqrt{\epsilon_{PEHE}}$ | $\epsilon_{ATE}$ |
| LR | $34.13_{\pm0.97}$ | $11.10_{\pm0.87}$ | $34.16_{\pm0.67}$ | $5.65_{\pm0.57}$ | $20.20_{\pm0.47}$ | $4.69_{\pm0.36}$ | $27.78_{\pm0.36}$ | $13.55_{\pm0.52}$ | $34.12_{\pm0.60}$ | $12.81_{\pm1.09}$ | $24.67_{\pm0.55}$ | $2.66_{\pm0.46}$ |
| TARNet | $32.25_{\pm2.92}$ | $8.73_{\pm2.83}$ | $30.75_{\pm1.83}$ | $3.84_{\pm2.30}$ | $19.96_{\pm1.10}$ | $3.63_{\pm1.26}$ | $26.13_{\pm0.79}$ | $14.73_{\pm0.56}$ | $30.45_{\pm1.21}$ | $14.57_{\pm0.83}$ | $21.38_{\pm0.93}$ | $2.96_{\pm0.74}$ |
| CFR$_{MMD}$ | $25.35_{\pm1.00}$ | $10.70_{\pm0.83}$ | $23.89_{\pm0.65}$ | $5.70_{\pm0.73}$ | $16.91_{\pm0.51}$ | $4.58_{\pm0.54}$ | $26.78_{\pm0.48}$ | $16.51_{\pm0.41}$ | $28.94_{\pm0.67}$ | $17.03_{\pm0.60}$ | $20.10_{\pm0.61}$ | $3.69_{\pm0.54}$ |
| CFR$_{Wass}$ | $37.26_{\pm5.94}$ | $10.46_{\pm6.63}$ | $49.70_{\pm9.04}$ | $12.37_{\pm7.83}$ | $23.33_{\pm4.05}$ | $6.57_{\pm3.88}$ | $25.14_{\pm1.81}$ | $13.62_{\pm2.64}$ | $33.93_{\pm5.36}$ | $14.31_{\pm6.11}$ | $25.12_{\pm2.38}$ | $2.94_{\pm2.36}$ |
| Netdeconf | $47.37_{\pm10.14}$ | $24.10_{\pm6.89}$ | $56.82_{\pm9.11}$ | $9.98_{\pm2.07}$ | $41.27_{\pm20.78}$ | $7.81_{\pm2.11}$ | $28.58_{\pm2.72}$ | $15.09_{\pm4.28}$ | $41.00_{\pm8.32}$ | $17.38_{\pm8.37}$ | $28.39_{\pm2.62}$ | $4.01_{\pm2.96}$ |
| GNN-HSIC | $34.00_{\pm3.87}$ | $8.98_{\pm5.09}$ | $36.31_{\pm4.58}$ | $6.23_{\pm3.88}$ | $21.63_{\pm2.39}$ | $5.46_{\pm3.96}$ | $32.93_{\pm3.32}$ | $11.77_{\pm4.86}$ | $50.44_{\pm7.14}$ | $16.41_{\pm8.30}$ | $35.97_{\pm4.56}$ | $6.20_{\pm3.38}$ |
| GCN-HSIC | $37.21_{\pm3.49}$ | $8.60_{\pm5.75}$ | $37.75_{\pm3.55}$ | $4.70_{\pm2.77}$ | $20.43_{\pm0.70}$ | $3.43_{\pm1.42}$ | $31.45_{\pm3.16}$ | $11.87_{\pm3.64}$ | $49.21_{\pm9.39}$ | $16.43_{\pm9.97}$ | $33.51_{\pm4.25}$ | $4.79_{\pm2.78}$ |
| NetEst | $32.94_{\pm5.87}$ | $14.39_{\pm6.95}$ | $29.93_{\pm4.26}$ | $7.32_{\pm6.66}$ | $19.53_{\pm1.23}$ | $4.75_{\pm4.01}$ | $29.66_{\pm3.16}$ | $11.12_{\pm2.28}$ | $36.87_{\pm2.60}$ | $13.11_{\pm2.25}$ | $26.34_{\pm3.44}$ | $4.15_{\pm1.89}$ |
| HyperSCI | $28.56_{\pm1.02}$ | $12.98_{\pm1.01}$ | $27.88_{\pm0.70}$ | $6.04_{\pm1.00}$ | $17.65_{\pm0.36}$ | $5.57_{\pm0.57}$ | $25.22_{\pm0.60}$ | $13.17_{\pm0.89}$ | $34.95_{\pm0.49}$ | $12.65_{\pm1.25}$ | $23.67_{\pm0.26}$ | $4.77_{\pm0.52}$ |
| IGL$_{GCN}$ | $\mathbf{23.03_{\pm0.27}}$ | $\mathbf{8.49_{\pm0.44}}$ | $\mathbf{22.57_{\pm0.12}}$ | $\mathbf{3.72_{\pm0.52}}$ | $\mathbf{15.62_{\pm0.08}}$ | $\mathbf{3.38_{\pm0.30}}$ | $23.43_{\pm0.92}$ | $12.67_{\pm1.61}$ | $26.18_{\pm1.12}$ | $11.72_{\pm2.63}$ | $18.82_{\pm0.22}$ | $\mathbf{2.56_{\pm1.79}}$ |
| IGL$_{GAT}$ | $23.88_{\pm0.30}$ | $10.35_{\pm0.60}$ | $22.99_{\pm0.52}$ | $5.26_{\pm0.50}$ | $16.00_{\pm0.23}$ | $4.18_{\pm0.27}$ | $\mathbf{22.82_{\pm0.73}}$ | $\mathbf{10.17_{\pm1.45}}$ | $\mathbf{24.88_{\pm0.92}}$ | $\mathbf{9.09_{\pm2.69}}$ | $\mathbf{18.80_{\pm0.25}}$ | $2.85_{\pm1.47}$ |

**Table 3: Performance comparison of ITE estimation over different values of $\lambda_s$ on Twitch-Explicit dataset.**

| | $\lambda_s = 1.0$ | | $\lambda_s = 3.0$ | | $\lambda_s = 5.0$ | |
|---|---|---|---|---|---|---|
| Method | $\sqrt{\epsilon_{PEHE}}$ | $\epsilon_{ATE}$ | $\sqrt{\epsilon_{PEHE}}$ | $\epsilon_{ATE}$ | $\sqrt{\epsilon_{PEHE}}$ | $\epsilon_{ATE}$ |
| LR | $29.49_{\pm0.70}$ | $7.14_{\pm0.60}$ | $30.31_{\pm0.88}$ | $7.25_{\pm0.79}$ | $32.65_{\pm0.97}$ | $7.30_{\pm0.72}$ |
| TARNet | $27.65_{\pm1.95}$ | $5.40_{\pm2.13}$ | $29.83_{\pm1.87}$ | $6.35_{\pm2.07}$ | $31.44_{\pm1.96}$ | $6.43_{\pm2.25}$ |
| GNN-HSIC | $30.64_{\pm3.61}$ | $6.89_{\pm4.31}$ | $33.87_{\pm3.63}$ | $6.99_{\pm3.31}$ | $36.43_{\pm2.62}$ | $7.82_{\pm3.23}$ |
| NetEst | $27.46_{\pm3.78}$ | $8.82_{\pm5.87}$ | $29.66_{\pm3.52}$ | $8.74_{\pm4.69}$ | $30.55_{\pm2.83}$ | $9.03_{\pm4.54}$ |
| HyperSCI | $24.69_{\pm0.69}$ | $8.19_{\pm0.86}$ | $26.44_{\pm1.26}$ | $8.97_{\pm1.35}$ | $27.97_{\pm1.24}$ | $9.12_{\pm1.40}$ |
| IGL (ours) | $20.41_{\pm0.16}$ | $5.20_{\pm0.42}$ | $21.30_{\pm0.43}$ | $5.30_{\pm0.59}$ | $22.44_{\pm0.57}$ | $5.75_{\pm0.87}$ |

## 4.3 Main Results

We begin by estimating the ITE and compare it with state-of-the-art methods. IGL employs two distinct GNN architectures as backbone GNN encoders: the GCN [25] and the GAT [48]. The results are presented in Table 2. These findings reveal that our proposed framework consistently surpasses all baselines. Such results underscore the effectiveness of IGL in handling graph data across varied domains, particularly in identifying invariant confounders and spillover effects across environments. Furthermore, an analysis of the baseline results indicates that several GNN-based methods do not necessarily outpace some of their traditional counterparts. To illustrate, within the Twitch-Explicit dataset's ES graph, both Netdeconf and GCN-HSIC yield results that are markedly inferior to those of TARNET and CFR$_{MMD}$. Similarly, for the Facebook-100 dataset's Penn graph, several techniques that emphasize graph structure, including GNN-HSIC, GCN-HISC, NetEst, and Netdeconf, underperform when compared to CFR and TARNet. These results suggest that excessive reliance on the training graph's structure by these methods can expose vulnerabilities when faced with variations in graph structures. In contrast, our approach demonstrates a robust capability to counteract such distribution shifts in graph structures, ensuring outstanding performance.

## 4.4 In-depth Analysis

To investigate the effects of structure distribution shifts on spillover effect modeling, we adjust the hyperparameter $\lambda_s$ to increase the significance of the spillover effect on the generated outcomes. For comparative analysis, we select both traditional estimators and those that account for the spillover effect on the graph as baseline

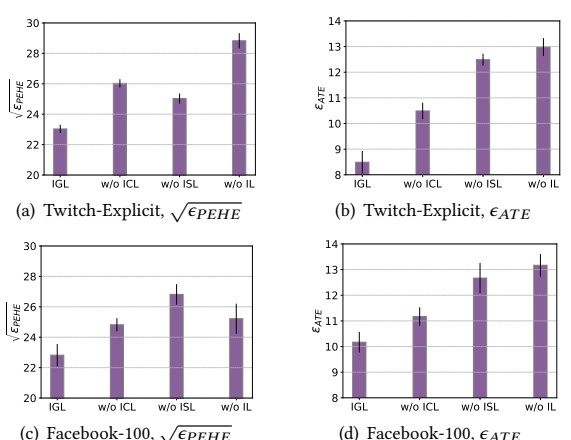

(a) Twitch-Explicit, $\sqrt{\epsilon_{PEHE}}$    (b) Twitch-Explicit, $\epsilon_{ATE}$

(c) Facebook-100, $\sqrt{\epsilon_{PEHE}}$    (d) Facebook-100, $\epsilon_{ATE}$

**Figure 3: Ablation studies of different components of our proposed IGL framework.**

methods. The experimental results are shown in Table 3. The results reveal that as the spillover effect's influence increases, there exists a consistent decline in the performance of both traditional and GNN-based methods. This uniformity in the performance downturn suggests that alterations in graph structure significantly impact the modeling of spillover effects. In comparison to the baselines, our approach yields superior performance gains. These results underscore our claim that distribution shifts in graph structure will magnify the spillover effect's impact on ITE estimation. Our method overcomes this issue by mastering the invariance of the spillover effect across varied graph structures.

## 4.5 Ablation Study

In the IGL framework, two pivotal components are present: invariant confounder learning and invariant spillover effect modeling. To ascertain the significance of these components on overall performance, we conduct experiments omitting each module separately. The results are depicted in Figure 3. Specifically, "w/o ICL" indicates the removal of the invariant learning specific to confounders, while "w/o ISL" refers to the absence of invariant learning for spillover effects. "w/o IL" represents a scenario where all invariant learning is eliminated; under this condition, $K$ environments are randomly generated and the loss function's mean and variance are directly

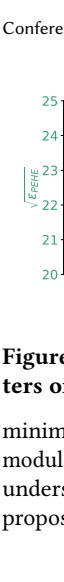
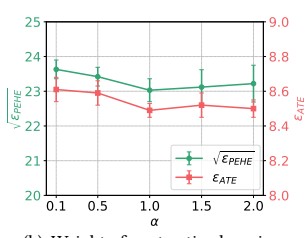
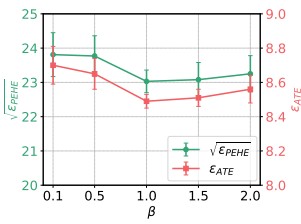
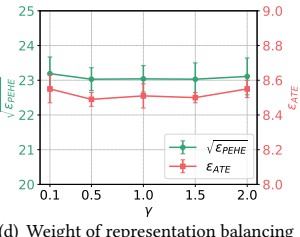

(a) Environment numbers    (b) Weight of contrastive learning    (c) Weight of invariant learning    (d) Weight of representation balancing

**Figure 4: ITE estimation performance (mean and standard error) of the proposed IGL framework under different hyperparameters on Twitch-Explicit dataset.**

minimized. It's evident from our results that omitting any single module leads to a substantial decrease in overall performance. This underscores the vital role and efficacy of each component in our proposed framework.

## 4.6 Sensitivity Analysis

To assess the sensitivity of IGL to hyperparameters, we evaluate the performance under various configurations. In particular, we set the number of environment generators with $K \in \{2, 3, 4, 5\}$, set weights for contrastive learning with $\alpha \in \{0.1, 0.5, 1.0, 1.5, 2.0\}$, for invariant learning with $\beta \in \{0.1, 0.5, 1.0, 1.5, 2.0\}$, and for representation balancing with $\gamma \in \{0.1, 0.5, 1.0, 1.5, 2.0\}$. From the results in Figure 4, it is evident that using more than three environments generally yields improved performance. Regarding the weights for contrastive and invariant learning, there is minimal perceptible variation in performance once the coefficient exceeds 1. Additionally, the model's final performance demonstrates robustness to fluctuations in representation balancing weights.

## 5 RELATED WORK

**Treatment Effect Estimation on Graphs.** While treatment effect estimation has achieved significant breakthroughs, the focus on graphs [32] has drawn great attention in recent years. Units on graphs or networked data are distinct from independent units due to their inherent correlations, presenting unique challenges for treatment effect estimation: i) Hidden confounders [41, 50]. The homophily [35] of network structures in graphs can introduce new confounders for causal effects beyond inherent unit features. ii) Spillover Effects or Interference [5, 6, 23, 45, 47, 57]. A unit's potential outcome is influenced not only by its own attributes but also by treatment assignments to neighboring units, implying a violation of the SUTVA assumption [40]. To address the challenge of hidden confounders, Netdeconf [18] employs the message-passing capability of the GCN [25] model; IGNITE [16] adopts the min-max game to resolve optimization conflict issues; GIAL [11] utilizes informax [49] to rectify imbalances in network structures; DNDC [31] is tailored for intricate dynamic graph scenarios. To mitigate the impact of spillover effects, [34] harnesses the message-passing mechanism of the GNN [25, 48] model to account for neighboring interference; NetEst [24] corrects the limitations of GNNs in ITE estimation through adversarial learning; HyperSCI [33] considers more complex hypergraph scenarios, which grapple with high-order interference challenges. While these techniques showcase commendable performance for singular graph data, they often overlook inductive

learning across multiple graphs in diverse environments, a scenario we argue is prevalent in real-world applications [19, 58].

**Invariant Learning on Graphs.** Invariant learning [1, 3, 8, 12, 38] aims to extract invariant relationships between observed data features and ground-truth outcomes across various environments, enhancing out-of-distribution (OOD) generalization ability. In the graph domain, invariant learning [10, 13, 26, 30, 43, 51, 52] has also emerged as a leading strategy for generalization. This pivot is rooted in the underlying assumption of data generation: the presence of stable features in graph data. These stable features, alternately termed "rationales" [30, 52] or "invariant features" [10] in existing literature, maintain an invariant relationship with ground-truth labels irrespective of the distribution. To capture these features, DIR [52] applies interventions on environmental features, while GREA [30] employs environment removal and replacement augmentations. To differentiate between stable and environmental features, both CAL [43] and DisC [13] variably adjust environmental feature representations, ensuring model predictions remain invariant across the environmental changes. The success of these studies demonstrates the power of invariant learning in handling distribution shifts in graph structures. In treatment effect estimation, invariant learning [42] also shows the potential ability to find appropriate confounders. Drawing inspiration from these pioneering efforts, our work ventures to learn invariant representations of confounders and spillover effects across diverse environments, thereby enhancing inductive learning for treatment effect estimation.

## 6 CONCLUSION

In this paper, we have delved into the challenge of estimating treatment effects on networked data across diverse environments, which is a scenario frequently encountered in real-world applications. Given the potential for graph data structures to evolve over time or vary across geographical locations, existing efforts for uncovering hidden confounders or modeling spillover effects may not be robust against such a problem. To counteract this vulnerability, we incorporate the invariance principle into treatment effect estimation, culminating in the development of the IGL framework. Specifically, we simulate varied environments by generating a multitude of new graphs. Then we prompt the model to derive invariant representations of confounders and spillover effects across these environments. This strategy ensures that our model becomes resilient to environmental shifts, paving the way for superior generalization capabilities when faced with unseen environments. Our comprehensive experiments shed light on the efficacy of the IGL framework, with results and detailed analyses affirming its advantages.

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
