# OpenReview forum: "Invariant Graph Learning for Treatment Effect Estimation from Networked Observational Data"
_ACM.org/TheWebConf/2024/Conference — TheWebConf24_

### Official Review · Reviewer_5J7d · 2023-11-14

**Novelty:** 4
**Technical Quality:** 4

**Review:**

This paper introduces the Invariant Graph Learning (IGL) framework for treatment effect estimation from networked observational data. The framework generates multiple networked data to simulate diverse environments and encourages the model to learn environment-invariant representations for confounders and spillover effects. The IGL model demonstrates superior performance compared to state-of-the-art methods on real-world datasets.

Pros:
1. The paper introduces the principle of invariance to the task of treatment effect estimation on networked data, which allows the model to extrapolate beyond a single observed environment.
2. The proposed Invariant Graph Learning (IGL) framework demonstrates superior performance compared to state-of-the-art methods in extensive experiments on two real-world datasets.

Cons:
1. The time complexity is huge. Generating new graph structures require quadratic complexity, which is unacceptable for large graphs.
2. Some important details are missing. For instance, GNN encoder $\Psi$ (mentioned in line 449) receives no gradients during training due to sampling operations. How do the authors address this issue?

**Questions:**

1. Can you provide more explanations or visualizations regarding which properties are invariant across different environments?
2. Why is the variance term added in Eq. (13)? Can the authors provide some theoretical justifications?
3. See Con 2.

**Ethics Review Description:**

Not applicable.

**Reviewer Confidence:**

3: The reviewer is confident but not certain that the evaluation is correct

**Scope:**

4: The work is relevant to the Web and to the track, and is of broad interest to the community

---

### Official Review · Reviewer_4NXp · 2023-11-21

**Novelty:** 5
**Technical Quality:** 5

**Review:**

This paper studies how to learn invariant graph representation learning model to estimate the treatment effect on graph-structutred data. More specifically, it proposes that confounders that are invariant to the perturbation of the graph structure (i.e., the environments mentioned in the paper) are critical to unbias estimation to the treatment effect. Therefore, it proposes a framework containing a perturbation generator and two graph learning models. It introduces an adversarial training framework, where the perturbation generator tries to modify the learnt representation to influence the result while the graph learning models try to minimize the influence from the generators. Empirical results show that the proposed model outperform the baselines on the benchmark dataset generated on real-world network structure with a simulator.

**Questions:**

1. I am curious about why we need also to apply the invariant learning on the spillover effect learning. The spillover effect actually relies on the network structure.

2. Could you provide a detailed justification to explain why the simulator is consistent with your theoretic assumption 2.1?

3. The invariant learning framework is very similar as adversarial training. So I am wondering, if we simply add this strategy on a normal GNN, will the performance of treatment estimation boost?

**Reviewer Confidence:**

3: The reviewer is confident but not certain that the evaluation is correct

**Scope:**

4: The work is relevant to the Web and to the track, and is of broad interest to the community

---

### Official Review · Reviewer_zbU9 · 2023-11-23

**Novelty:** 3
**Technical Quality:** 4

**Review:**

This paper introduces the Invariant Graph Learning (IGL) framework for treatment effect estimation from networked observational data. The framework generates multiple networked data to simulate diverse environments and encourages the model to learn environment-invariant representations for confounders and spillover effects. The IGL model demonstrates superior performance compared to state-of-the-art methods on real-world datasets.
Pros:
1. This paper is well-presented, with clear and sufficient description of the background, related works, the proposed method, and extensive experiments.
2. This paper delve into the critical importance of identifying invariant confounders and spillover effects for the task of treatment effect estimation on networked data, which helps the estimators achieve better inductive learning and generalize to diverse domains.
3. Extensive experiments validate the effectiveness of the proposed IGL framework on two real-world networked data. Detailed comparisons and in-depth analysis also further confirm the superiority of IGL.
Cons:
1. Limited contribution. The main approach of IGL is very similar to the IL approach based on contrast learning and data augmentation method with inadequate contributions. For example, GraphCL [1], DGCL [2], etc.
2. Insufficient experiments. Graph invariant learning methods are missing from the baselines, including but not limited to GraphCL [1] and DGCL [2]. In addition, the main experiment should not be limited to ITE estimation estimates, but should also consider the performance of relevant downstream tasks on the dataset, such as node classification, link prediction, etc. on the web data.

[1] You, Yuning, et al. "Graph contrastive learning with augmentations." Advances in neural information processing systems 33 (2020): 5812-5823.
[2] Li, Haoyang, et al. "Disentangled contrastive learning on graphs." Advances in Neural Information Processing Systems 34 (2021): 21872-21884.

**Questions:**

1. Have you ever considered change the name of your framework? As there already exists a pioneer graph invariant learning method GIL [3], which may make confusion.
2. What is the time and space complexity of IGL? How does it compare to baseline?
3. What are the biggest technical differences and advantages of IGL compared to GraphCL [1]? What is the most innovative contribution of IGL compared to other related methods?
[3] Li, Haoyang, et al. "Learning invariant graph representations for out-of-distribution generalization." Advances in Neural Information Processing Systems 35 (2022): 11828-11841.

**Reviewer Confidence:**

4: The reviewer is certain that the evaluation is correct and very familiar with the relevant literature

**Scope:**

4: The work is relevant to the Web and to the track, and is of broad interest to the community

---

### Official Review · Reviewer_2UJW · 2023-11-24

**Novelty:** 5
**Technical Quality:** 6

**Review:**

The paper is written well; the technical description is clear.

Overall, the paper seems to be technically solid.

The paper compares the performance with the several recent baselines and seems to outperform the baselines.

There are not many things to complain, but here are several minor suggestions that might improve the presentation of the paper:

- Figure 1 does not seem to provide much value. Rather it just increases the complexity of understanding.

- Although the paper provides neat presentation in terms of mathematics, simple ones are not defined such as ||.

- In Algorithm 1, it is better to give references to the main text equations.

**Questions:**

Please see the comments above.

**Reviewer Confidence:**

3: The reviewer is confident but not certain that the evaluation is correct

**Scope:**

4: The work is relevant to the Web and to the track, and is of broad interest to the community

---

### Decision · Program_Chairs · 2024-01-22

**Decision:**

Accept

**Comment:**

The reviewers appreciate the presentation of the paper as well as its technical contribution. Several technical questions were raised, which I found to have been adequately addressed by the authors during the rebuttal phase.

 The most notable point to remain was reviewer zbU9's concerns regarding novelty. A lengthy discussion ensued, which unfortunately escalated by taking on a rather unprofessional tone. I strongly advise the authors as well as the reviewer to avoid behaving this way in the future. No one wins this way.

 I ultimately believe that the paper can be expected. I'd lean towards accepting a paper whose novelty is not fully evident to everybody, rather than rejecting a paper that makes a solid contribution.